# Gaussian Process Conditional Density Estimation

**Vincent Dutordoir**[* 1]     **Hugh Salimbeni**[* 1,2]     **Marc Peter Deisenroth**[1,2]     **James Hensman**[1]

[1]PROWLER.io, Cambridge, UK    [2]Imperial College London

{vincent, hugh, marc, james}@prowler.io

## Abstract

Conditional Density Estimation (CDE) models deal with estimating conditional distributions. The conditions imposed on the distribution are the inputs of the model. CDE is a challenging task as there is a fundamental trade-off between model complexity, representational capacity and overfitting. In this work, we propose to extend the model's input with latent variables and use Gaussian processes (GPs) to map this augmented input onto samples from the conditional distribution. Our Bayesian approach allows for the modeling of small datasets, but we also provide the machinery for it to be applied to big data using stochastic variational inference. Our approach can be used to model densities even in sparse data regions, and allows for sharing learned structure between conditions. We illustrate the effectiveness and wide-reaching applicability of our model on a variety of real-world problems, such as spatio-temporal density estimation of taxi drop-offs, non-Gaussian noise modeling, and few-shot learning on omniglot images.

## 1   Introduction

Conditional Density Estimation (CDE) is the very general task of inferring the probability distribution $p(f(\mathbf{x}) \,|\, \mathbf{x})$, where $f(\mathbf{x})$ is a random variable for each $\mathbf{x}$. Regression can be considered a CDE problem, although the emphasis is on modeling the mapping rather than the conditional density. The conditional density is commonly Gaussian with parameters that depend on $\mathbf{x}$. This simple model for data may be inappropriate if the conditional density is multi-modal or has non-linear associations.

Throughout this paper we consider an input $\mathbf{x}$ to be the *condition*, and the output $\mathbf{y}$ to be a sample from the conditional density imposed by $\mathbf{x}$. For example, in the case of estimating the density of taxi drop-offs, the input or condition $\mathbf{x}$ could be the pick-up location and the output $\mathbf{y}$ would be the corresponding drop-off. In this context, we are more interested in learning the complete density over drop-offs rather than only a single point estimate, as we would expect the taxi drop-off to be multi-modal because passengers need to go to different places (e.g., airport/city center/suburbs). We would also expect the drop-off location to depend on the starting point and time of day: therefore, we are interested in *conditional* densities. In the experiment section we will return to this example.

In this work, we present a Gaussian process (GP) based model for estimating conditional densities, abbreviated as GP-CDE. While a vanilla GP used directly is unlikely to be a good model for conditional density estimation as the marginals are Gaussian, we extend the inputs to the model with latent variables to allow for modeling richer, non-Gaussian densities when marginalizing the latent variable. Fig. 1 shows a high-level overview of the model. The added latent variables are denoted by $\mathbf{w}$. The latent variable $\mathbf{w}$ and condition $\mathbf{x}$ are used as input of the GP. A *recognition/encoder* network is used to amortize the learning of the variational posterior of the latent variables. The matrices $\mathbf{A}$ and $\mathbf{P}$ act as probabilistic linear transforms on the input and the output of the GP, respectively.

The GP-CDE model is closely related to both supervised and unsupervised, non-Bayesian and Bayesian models. We first consider the relationship to parametric models, in particular Variational

---

[*]Equal contribution

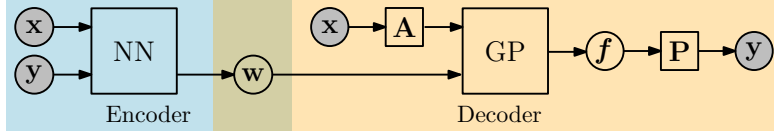

Figure 1: Diagram of the GP-Conditional Density Estimator. The GP-CDE consists of an encoder (blue) and a decoder (orange). The observed variables $\mathbf{x}$ and $\mathbf{y}$ are respectively the condition on the distribution and a sample from the conditional distribution. The encoder consists of a neural network, and uses the variables $\mathbf{x}$ and $\mathbf{y}$ as inputs to produce the parameters for the posterior of the latent variable $\mathbf{w}$. The decoder part is built out of a GP and two linear transformation matrices, $\mathbf{A}$ and $\mathbf{P}$. $\mathbf{A}$ is applied to the condition $\mathbf{x}$ to reduce the dimensionality of $\mathbf{x}$ before it is combined with the latent $\mathbf{w}$ and fed into the GP. The matrix $\mathbf{P}$ can be used to correlate the outputs of the latent function $\mathbf{f}$.

Autoencoders (VAEs) [17, 23] and their conditional counterparts (CVAE) [18, 24]. (C)VAEs use a deterministic, parametrized neural network as decoder, whereas we adopt a Bayesian non-parametric GP. Using a GP for this non-linear decoder mapping offers two advantages. First, it allows us to specify our prior beliefs about the mapping, resulting in a model that can gracefully accommodate sparse or missing data. Note that even large datasets can be sparse, e.g., the omniglot images or taxi drop-off locations at the fringes of a city. As the CVAE has no prior on the decoder mapping, the model can overfit the training data and the latent variable posterior becomes over-concentrated, leading to poor test log-likelihoods. Second, the GP allows tractable uncertainty propagation through the decoder mapping (for certain kernels). This allows us to calculate the variational objective deterministically, and we can further exploit natural gradients for fast inference. Neural network decoders do not admit such structure, and are typically optimized with general-purpose tools.

A second perspective can be seen through the connection to Gaussian process models. By dropping the latent variables $\mathbf{w}$ we recover standard multiple-output GP regression with sparse variational inference. If we drop the known inputs $\mathbf{x}$ and use only the latent variables, we obtain the Bayesian GP-LVM [26, 20]. Bayesian GP-LVMs are typically used for modeling complex distributions and non-linear mappings from a lower-dimensional latent variable into a high-dimensional space. By combining the GP-LVM framework with known inputs we create a model that outputs conditional samples in this high-dimensional space.

Our primary contribution is to show that our GP-CDE can be applied to a wide variety of settings, without the necessity for fine-tuning or regularization. We show that GP-CDE outperforms GP regression on regression benchmarks; we study the importance of accurate density estimation in high-dimensional spaces; and we deal with learning the correlations between conditions in a large spatio-temporal dataset. We achieve this through three specific contributions. (i) We extend the model of Wang and Neal [29] with linear transformations for the inputs and outputs. This allows us to deal with high-dimensional conditions and enables a priori correlations in the output. (ii) We apply natural gradients to address the difficulty of mini-batched optimization in [26]. (iii) We derive a free-form optimal posterior distribution over the latent variables. This provides a tighter bound and reduces the number of variational parameters to optimize.

## 2 Background: Gaussian Processes and Latent Variable Models

**Gaussian Processes** A Gaussian process (GP) is a Bayesian non-parametric model for functions. Bayesian models have two significant advantages that we exploit in this work: we can specify prior beliefs leading to greater data efficiency, and we can obtain uncertainty estimates for predictions. A GP is defined as a set of random variables $\{f(\mathbf{x}_1), f(\mathbf{x}_2), \ldots\}$, any finite subset of which follow a multivariate Gaussian distribution [22]. When a stochastic function $f : \mathbb{R}^D \rightarrow \mathbb{R}$ follows a GP it is fully specified by its mean $m(\cdot)$ and covariance function $k(\cdot, \cdot)$, and we write $f \sim \mathcal{GP}(m(\cdot), k(\cdot, \cdot))$.

The most common use of a GP is the regression task of inferring an unknown function $f$, given a set of $N$ observations $\mathbf{y} = [y_1, \ldots, y_N]^\top$ and corresponding inputs $\mathbf{x}_1, \ldots, \mathbf{x}_n$. The likelihood $p(y_i \mid f)$ generally is taken to depend on $f(\mathbf{x}_i)$ only, and the Gaussian likelihood $\mathcal{N}(y_n \mid f(\mathbf{x}_n), \sigma^2)$ is widely used as it results in analytical closed-form inference.

**Conditional Deep Latent Variable Models**   Conditional Deep Latent Variable Models (C-DLVMs) consist of two components: a prior distribution $p(\mathbf{w}_n)$ over the latent variables[2] which is assumed to factorize over the data, and a generator or decoder function $\mathbf{g}_\theta(\mathbf{x}_n, \mathbf{w}_n) : \mathbb{R}^{D_\mathbf{x} + D_\mathbf{w}} \to \mathbb{R}^{D_\mathbf{y}}$. The VAE and CVAE are examples where the generator function is a deep (convolutional) neural network with weights $\boldsymbol{\theta}$ [17, 24]. The outputs of the generator function are the parameters of a likelihood, commonly the Gaussian for continuous data or the Bernoulli for binary data. The joint distribution for a single data point is $p_\theta(\mathbf{y}_n, \mathbf{w}_n \,|\, \mathbf{x}_n) = p(\mathbf{w}_n) p_\theta(\mathbf{y}_n \,|\, \mathbf{x}_n, \mathbf{w}_n)$. We assume the data to be i.i.d., then the the marginal likelihood in the Gaussian case is given by

$$\log p_\theta(\mathbf{Y} \,|\, \mathbf{X}) = \sum\nolimits_n \log p_\theta(\mathbf{y}_n \,|\, \mathbf{x}_n) = \sum\nolimits_n \log \int p(\mathbf{w}_n) \, \mathcal{N}(\mathbf{y}_n \,|\, \mathbf{g}_\theta(\mathbf{x}_n, \mathbf{w}_n), \sigma^2 \mathbf{I}) \, \mathrm{d}\mathbf{w}_n \,, \quad (1)$$

where $\mathbf{X} = \{\mathbf{x}\}_{n=1}^N$, and likewise for $\mathbf{Y}$ and $\mathbf{W}$. As $\mathbf{g}_\theta(\mathbf{x}_n, \mathbf{w}_n)$ is a complicated non-linear function of its inputs, this integral cannot be calculated in closed-form. Kingma and Welling [17] and Rezende et al. [23] addressed this problem by using variational inference. Variational inference posits an approximate posterior distribution $q_\phi(\mathbf{W})$, and finds the closest $q_\phi$ to the true posterior, measured by KL divergence, i.e. $\arg\min_{q_\phi} \mathrm{KL}\left[q_\phi(\mathbf{W}) \| p(\mathbf{W} \,|\, \mathbf{X}, \mathbf{Y})\right]$. It can be shown that this optimization objective is equal to the Evidence Lower Bound (ELBO)

$$\log p_\theta(\mathbf{Y} \,|\, \mathbf{X}) \geq \mathcal{L} := \sum\nolimits_n \mathbb{E}_{q_\phi(\mathbf{w}_n)} \left[\log p_\theta(\mathbf{y}_n \,|\, \mathbf{x}_n, \mathbf{w}_n)\right] - \mathrm{KL}\left[q_\phi(\mathbf{w}_n) \| p(\mathbf{w}_n)\right].$$

A mean-field distribution is typically used for the latent variables $\mathbf{W}$ with a multivariate Gaussian form for $q_\phi(\mathbf{w}_n) = \mathcal{N}(\mathbf{w}_n \,|\, \boldsymbol{\mu}_{\mathbf{w}_n}, \boldsymbol{\Sigma}_{\mathbf{w}_n})$. Rather than representing the Gaussian parameters $\boldsymbol{\mu}_{\mathbf{w}_n}$ and $\boldsymbol{\Sigma}_{\mathbf{w}_n}$ for each data point directly, Kingma and Welling [17] and Rezende et al. [23] instead amortize these parameters into a set of global parameters $\phi$, where $\phi$ parameterizes an auxiliary function $\mathbf{h}_\phi : (\mathbf{x}_n, \mathbf{y}_n) \mapsto (\boldsymbol{\mu}_{\mathbf{w}_n}, \boldsymbol{\Sigma}_{\mathbf{w}_n})$, referred to as the encoder/recognition network.

## 3   Conditional Density Estimation with Gaussian Processes

This section details our model and the inference scheme. Our contributions are threefold:(i) we derive an optimal free-form variational distribution $q(\mathbf{W})$ (Section 3.2); (ii) we ease the burden of jointly optimizing $q(\boldsymbol{f}(\cdot))$ and $q(\mathbf{W})$ using natural gradients (Section 3.2.1) for the variational parameters of $q(\boldsymbol{f}(\cdot))$; (iii) we extend the model to allow for the modeling of high-dimensional inputs and impose correlation on the outputs using linear transformations (Section 3.3).

### 3.1   Model

The key idea of our model is to substitute the neural network decoder in the C-DLVM framework with a GP, see Fig. 1. Treating the decoder in a Bayesian manner leads to several advantages. In particular, in the small-data regime, a probabilistic decoder will be advantageous to leverage prior assumptions and avoid over-fitting.

As we want to apply our model on both high-dimensional correlated outputs (e.g., images) and high-dimensional inputs (e.g., one-hot encodings of omniglot labels), we introduce two matrices $\mathbf{A}$ and $\mathbf{P}$. They are used for probabilistic linear transformations of the inputs and the outputs, respectively. The likelihood of the GP-CDE model is then given by

$$p_\theta(\mathbf{y}_n \,|\, \mathbf{x}_n, \mathbf{w}_n, \boldsymbol{f}(\cdot), \mathbf{A}, \mathbf{P}) = \mathcal{N}\left(\mathbf{y}_n \,\big|\, \mathbf{P}\boldsymbol{f}\big([\mathbf{A}\mathbf{x}_n, \mathbf{w}_n]\big), \sigma^2 \mathbf{I}\right),$$

where $[\cdot, \cdot]$ denotes concatenation. We assume the GP $\boldsymbol{f}(\cdot)$ consists of $L$ independent GPs $f_\ell(\cdot)$ for each output dimension $\ell$. The latent variables are a priori independent for each data point and have a standard-normal prior distribution. We discuss priors for $\mathbf{A}$ and $\mathbf{P}$ in section 3.3.

### 3.2   Inference

In this section, we present our inference scheme, initially in the case without $\mathbf{A}$ and $\mathbf{P}$ to lighten the notation. We will return to these matrices in section section 3.3. We calculate an ELBO on the marginal

likelihood, similarly to (1). Assuming a factorized posterior $q(\boldsymbol{f}(\cdot), \mathbf{W}) = q(\boldsymbol{f}(\cdot)) \prod_n q(\mathbf{w}_n)$, where $\mathbf{W} = \{\mathbf{w}_n\}_{n=1}^N$, between the GP and the latent variables we get the ELBO

$$\mathcal{L} = \sum_n \left\{ \mathbb{E}_{q(\mathbf{w}_n)} \mathbb{E}_{q(\boldsymbol{f}(\cdot))} \left[ \log p(\mathbf{y}_n \mid \boldsymbol{f}(\cdot), \mathbf{x}_n, \mathbf{w}_n) \right] - \mathrm{KL}\left[ q(\mathbf{w}_n) \| p(\mathbf{w}_n) \right] \right\}$$
$$- \mathrm{KL}\left[ q(\boldsymbol{f}(\cdot)) \| p(\boldsymbol{f}(\cdot)) \right]. \quad (2)$$

Since the ELBO is a sum over the data we can calculate unbiased estimates of the bound using mini-batches. We follow Hensman et al. [15] and choose independent sparse GPs over the output dimensions $q(f_\ell(\cdot)) = \int p(f_\ell(\cdot) \mid \mathbf{u}_\ell) q(\mathbf{u}_\ell) \, \mathrm{d}\mathbf{u}_\ell$, where $\ell = 1, \dots, L$ and $\mathbf{u}_\ell \in \mathbb{R}^M$ are inducing outputs corresponding to the inducing inputs $\mathbf{z}_m$, $m = 1, \dots, M$, so that $u_{ml} = f_\ell(z_m)$. We choose $q(\mathbf{u}_\ell) = \mathcal{N}(\mathbf{u}_\ell \mid \mathbf{m}_\ell, \mathbf{S}_\ell)$. Since $p(f_\ell(\cdot) \mid \mathbf{u}_\ell)$ is conjugate to $q(\mathbf{u}_\ell)$, the integral can be calculated in closed-form. The result is a new sparse GP for each of the output dimensions $q(f_\ell(\cdot)) = \mathcal{GP}(\mu_\ell(\cdot), \sigma_\ell(\cdot, \cdot))$ with closed-form mean and variance. See Appendix C for a detailed derivation. Using the results of Matthews et al. [21], the KL-term over the multi-dimensional latent function $\boldsymbol{f}(\cdot)$ simplifies to $\sum_\ell \mathrm{KL}\left[ q(\mathbf{u}_\ell) \| p(\mathbf{u}_\ell) \right]$, which is closed-form, since $q(\mathbf{u}_\ell)$ and $p(\mathbf{u}_\ell)$ are both Gaussian.

The inner expectation over the variational posterior $q(\boldsymbol{f}(\cdot))$ in (2) can be calculated in closed-form (see Appendix C) as the likelihood is Gaussian. We define this analytically tractable quantity as

$$\mathcal{L}_{\mathbf{w}_n} = \mathbb{E}_{q(\boldsymbol{f}(\cdot))} \left[ \log p(\mathbf{y}_n \mid \boldsymbol{f}(\cdot), \mathbf{x}_n, \mathbf{w}_n) \right]. \quad (3)$$

Using this definition, and using the sparse variational posterior for $q(\boldsymbol{f}(\cdot))$ as described above, we write the bound in (2) as

$$\mathcal{L} = \sum_n \left\{ \mathbb{E}_{q(\mathbf{w}_n)} \mathcal{L}_{\mathbf{w}_n} - \mathrm{KL}\left[ q(\mathbf{w}_n) \| p(\mathbf{w}_n) \right] \right\} - \sum_\ell \mathrm{KL}\left[ q(\mathbf{u}_\ell) \| p(\mathbf{u}_\ell) \right]. \quad (4)$$

We consider two options for $q(\mathbf{w}_n)$: (i) we can either make a further Gaussian assumption, and have a variational posterior of the form $q(\mathbf{w}_n) = \mathcal{N}(\mathbf{w}_n \mid \boldsymbol{\mu}_{\mathbf{w}_n}, \boldsymbol{\Sigma}_{\mathbf{w}_n})$, or (ii) we can find the analytically optimal value of the bound for a free-form $q(\mathbf{w}_n)$.

**(i) Gaussian** $q(\mathbf{w}_n)$ First, a Gaussian $q(\mathbf{w}_n)$ implies that the KL over the latent variables is closed-form, as both the prior and posterior are Gaussian. Therefore, we are left with the calculation of the first term in (4) $\mathbb{E}_{q(\mathbf{w}_n)} \mathcal{L}_{\mathbf{w}_n}$. We follow the approach of C-DLVMs, explained in Section 2, and use Monte Carlo sampling to estimate the expectation. To enable differentiability, we use the re-parameterization trick and write $\mathbf{w}_n = \boldsymbol{\mu}_{\mathbf{w}_n} + \mathbf{L}_{\mathbf{w}_n} \boldsymbol{\xi}_n$ with $p(\boldsymbol{\xi}_n) = \mathcal{N}(\mathbf{0}, \mathbf{I})$, independent for each data point, and $\mathbf{L}_{\mathbf{w}_n} \mathbf{L}_{\mathbf{w}_n}^\top = \boldsymbol{\Sigma}_{\mathbf{w}_n}$. Note that this does not change the distribution of $q(\mathbf{w}_n)$, but now the expectation is over a parameterless distribution. We can then take a differentiable unbiased estimate of the bound by sampling from $\boldsymbol{\xi}_n$.

In practice, rather than represent the Gaussian parameters $\boldsymbol{\mu}_{\mathbf{w}_n}$ and $\mathbf{L}_{\mathbf{w}_n}$ for each data-point directly, we instead amortize these parameters into a set of global parameters $\phi$, where $\phi$ parameterizes an auxiliary function $\mathbf{h}_\phi$, (or 'recognition network') of the data: $(\boldsymbol{\mu}_{\mathbf{w}_n}, \mathbf{L}_{\mathbf{w}_n}) = \mathbf{h}_\phi(\mathbf{x}_n, \mathbf{y}_n)$. This is identical to the decoder component of C-DLVMs.

An alternative approach would be to use the kernel expectation results of Girard et al. [13] to evaluate $\mathbb{E}_{q(\mathbf{w}_n)} \mathcal{L}_{\mathbf{w}_n}$. Using these results we can evaluate the bound in closed-form, rather than an approximate using Monte Carlo. However, the computations involved in calculating the kernel expectations can be prohibitive, as it requires evaluating a $NM^2 D_{\mathbf{y}}$ sized tensor. Furthermore, closed-form solutions for the kernel expectations only exist for RBF and polynomial kernels, which makes this approach less favorable in practice.

**(ii) Analytically optimal** $q(\mathbf{w}_n)$ So far, we assumed that the variational distribution $q(\mathbf{w}_n)$ is Gaussian. When $q(\cdot)$ is non-Gaussian, it is possible to integrate over $\mathbf{w}_n$ with quadrature as we detail in the following. We first bound the conditional $p(\mathbf{Y} \mid \mathbf{X}, \mathbf{W})$ and use the same sparse variational posterior for the GP as before, to obtain

$$\log p(\mathbf{Y} \mid \mathbf{X}, \mathbf{W}) \geq \sum_n \mathcal{L}_{\mathbf{w}_n} - \sum_\ell \mathrm{KL}\left[ q(\mathbf{u}_\ell) \| p(\mathbf{u}_\ell) \right].$$

As shown in [15] and explained above we can calculate $\mathcal{L}_{\mathbf{w}_n}$ analytically. By expressing the marginal likelihood as $\log p(\mathbf{Y}\,|\,\mathbf{X}) = \log \int p(\mathbf{Y}\,|\,\mathbf{X},\mathbf{W})p(\mathbf{W})\,\mathrm{d}\mathbf{W}$, we get

$$\log p(\mathbf{Y}\,|\,\mathbf{X}) \geq \log \int \exp\left(\sum_n \mathcal{L}_{\mathbf{w}_n} - \sum_\ell \mathrm{KL}\left[q(\mathbf{u}_\ell)\|p(\mathbf{u}_\ell)\right]\right) p(\mathbf{W})\,\mathrm{d}\mathbf{W}$$

$$= \sum_n \log \int \exp\left(\mathcal{L}_{\mathbf{w}_n}\right) p(\mathbf{w}_n)\,\mathrm{d}\mathbf{w}_n - \sum_\ell \mathrm{KL}\left[q(\mathbf{u}_\ell)\|p(\mathbf{u}_\ell)\right]$$

where we exploited the monotonicity of the logarithm. We can compute this integral with quadrature when $\mathbf{w}_n$ is low-dimensional (the dimensionality of $\mathbf{x}_n$ does not matter here). Assuming we have sufficient quadrature points, this gives the *analytically optimal* bound for $q(\mathbf{w}_n)$. The analytical optimal approach does not resort to the Gaussian approximation for $q(\mathbf{w}_n)$, so it is a tighter bound. See Appendix D for a proof that this bound is necessarily tighter than the bound in the Gaussian case.

### 3.2.1 Natural Gradient

Optimizing $q(\mathbf{u}_\ell)$ together with $q(\mathbf{W})$ can be challenging due to problems of local optima and the strong coupling between the inducing outputs $\mathbf{u}_\ell$ and the latent variables $\mathbf{W}$. One option is to analytically optimize the bound with respect to the variational parameters of $\mathbf{u}_\ell$, but this prohibits the use of mini-batches and reduces the applicability to large-scale problems. Recall that the variational parameters of $\mathbf{u}_\ell$ are the mean and the covariance of the approximate posterior distribution $q(\mathbf{u}_\ell) = \mathcal{N}(\mathbf{m}_\ell, \mathbf{S}_\ell)$ over the inducing outputs. We can use the natural gradient [3] to update the variational parameters $\mathbf{m}_\ell$ and $\mathbf{S}_\ell$.

This approach has the attractive property of recovering exactly the analytically optimal solution for $q(\mathbf{u}_\ell)$ in the full-batch case if the natural gradient step size is taken to be 1. While natural gradients have been used before in GP models [15], they have not been used in combination with uncertain inputs. Due to the quadratic form of the log-likelihood as a function of the kernel inputs $\mathbf{X}, \mathbf{W}$ and $\mathbf{Z}$, we can calculate the expectation w.r.t. $q(\mathbf{W})$, which will still be quadratic in the inducing outputs $\mathbf{u}_\ell$. Therefore, the expression is still conjugate, and the natural gradient step of size 1 recovers the analytic solution.

In practice, the natural gradient is used for the Gaussian variational parameters $\mathbf{m}_\ell$ and $\mathbf{S}_\ell$, and ordinary gradients are used for the inducing inputs $\mathbf{Z}$, the recognition network (if applicable) and other hyperparameters of the model (the kernel and likelihood parameters). The variational parameters of $q(\mathbf{A})$ and the parameters of $\mathbf{P}$ are also updated using the ordinary gradient.

### 3.3 Probabilistic Linear Transformations

**Input** For high-dimensional inputs it may not be appropriate to define a GP directly in the augmented space $[\mathbf{x}_n, \mathbf{w}_n] \in \mathbb{R}^{D_\mathbf{x} + D_\mathbf{w}}$. This might be the case if the input data is a one-hot encoding of many classes. We can extend our model with a linear projection to a lower-dimensional space before concatenating with the latent variables, $[\mathbf{A}\mathbf{x}_n, \mathbf{w}_n]$. We denote this projection matrix by $\mathbf{A}$, as shown in Fig. 1.

We use an isotropic Gaussian prior for elements of $\mathbf{A}$ and a Gaussian variational posterior that factorizes between $\mathbf{A}$ and the other variables in the model: $q(\boldsymbol{f}(\cdot), \mathbf{W}, \mathbf{A}) = q(\boldsymbol{f}(\cdot))q(\mathbf{W})q(\mathbf{A})$. For Gaussian $q(\mathbf{w})$ the bound is identical as in (2) except that we include an additional $-\mathrm{KL}\left[q(\mathbf{A})\|p(\mathbf{A})\right]$ term and include the mean and variance for $\mathbf{A}\mathbf{x}$ as the input of the GP. A similar approach was used in the regression case by Titsias and Lázaro-Gredilla [27].

**Output** We can move beyond the assumption of a priori independent outputs to a correlated model by using a linear transformation of the outputs of the GP. This model is equivalent to a 'multi-output' GP model with a linear model of covariance between tasks. In the multi-output GP framework [2], the $D_\mathbf{y}$ outputs are stacked to a single vector of length $ND_\mathbf{y}$, and a single GP is used jointly with a structured covariance. In the simplest case, the covariance can be structured as $\mathbf{R} \otimes \mathbf{K}$, where $\mathbf{R}$ can be any positive semi-definite matrix of size $D_\mathbf{y} \times D_\mathbf{y}$, and $\mathbf{K}$ is an $N \times N$ matrix. By transforming the outputs with the matrix $\mathbf{P}$ we recover exactly this model with $\mathbf{R} = \mathbf{P}^\top \mathbf{P}$. Apart from the simplicity of implementation, another advantage is that we can handle degenerate cases (i.e., where the number of outputs is less than $D_\mathbf{y}$) without having to deal with issues of ill-conditioning. It would be possible to use a Gaussian prior for $\mathbf{P}$ while retaining conjugacy, but in our experiments we use a non-probabilistic $\mathbf{P}$ and optimize it using MAP.

# 4 Related work

The GP-CDE model is closely related to both supervised an unsupervised Gaussian process based models. If we drop the latent variables $\mathbf{W}$ our approach recovers standard multiple-output Gaussian process regression with sparse variational inference. If we drop the known inputs $\mathbf{X}$ and use only the latent variables, we obtain a Bayesian GP-LVM [26]. Bayesian GP-LVMs are typically used for modeling complex distributions and non-linear mappings from a lower-dimensional variable into a high-dimensional space. By combining the GP-LVM framework with known inputs we create a model that outputs conditional samples in this high-dimensional space. Differently from Titsias and Lawrence 26, during inference we do not marginalize out the inducing variables $\mathbf{u}_\ell$ but rather treat them as variational parameters of the model. This scales our model to arbitrarily large datasets through the use of Stochastic Variational Inference (SVI) [15, 16]. While the mini-batch extension to the Bayesian GP-LVM was suggested in [15], its notable absence from the literature may be due to the difficulty in the joint optimization of $\mathbf{W}$ and $\mathbf{u}_\ell$. We found that the natural gradients were essential to alleviate this problem. A comparison demonstrating this is presented in the experiments.

Wang and Neal [29] proposed the Gaussian Process Latent Variable model (GP-LV), which is a special case of our model. The inference they employ is based on Metroplis sampling schemes and does not scale to large datasets or high dimensions. In this work, we extend their model using linear projection matrices on both input and output, and we present an alternative method of inference that scales to large datasets. Damianou and Lawrence [11] also propose a special case of our model, though they use it for missing data imputation rather than to induce non-Gaussian densities. They also use sparse variational inference, but they analytically optimize $q(\mathbf{u}_\ell)$ so cannot use mini-batches. Depeweg et al. [12] propose a similar model, but use a Bayesian neural network instead of a GP.

The use of a recognition model, as in VAEs, was first proposed by Lawrence and Quiñonero-Candela [20] in the context of a GP-LVM, though it was motivated as a constraint on the latent variable locations rather than an amortization of the optimization cost. Recognitions models were later used by Bui and Turner [7] and by Dai et al. [9] for deep GPs.

A GP model with latent variables and correlated multiple outputs was recently proposed in Dai et al. [10]. In this model, the latent variables determine the correlations between outputs via a Kronecker-structured covariance, whereas we have a fixed between-output covariance. That is, in our model the covariance of the stacked outputs is $(\mathbf{P}\mathbf{P}^\top) \otimes (\mathbf{K_X K_W})$, whereas in Dai et al. [10] the covariance is $\mathbf{K_W} \otimes \mathbf{K_X}$. These models are complementary and perform different functions. [6] proposed a model that is also similar to ours, but with categorical variables in the latent space. Other approaches to non-parametric density estimation include modeling the log density directly with a GP [1], and using an infinite generalization of the exponential family [25] which was recently extended to the conditional case [4].

# 5 Experiments

**Large-scale spatio-temporal density estimation** We apply our model to a New York City taxi dataset to perform conditional spatial density estimation. The dataset holds records of more than $1.4$ million taxi trips, which we filter to include trips that start and end within the Manhattan area. Our objective is to predict spatial distributions of the drop-off location, based on the pick-up location, the day of the week, and the time of day. The two temporal features are encoded as sine and cosine with the natural periods, giving 6-dimensional inputs in total[3]. Trippe and Turner [28] follow a similar setup to predict a distribution over the pick-up locations given the fare and the tip of the ride.

Table 1 compares the performance of 6 different models, unconditional and conditional Kernel Density Estimation (U-KDE, C-KDE), Mixture Density Networks (MDN-$k$, $k = 1, 5, 10, 50$) [5], our GP-CDE model, a simple GP model and the unconditional GP-LVM [26]. We evaluate the models using negative log predictive probability (NLPP) of the test set. The test sets are constructed by sequentially adding points that have greatest minimum distance from the testing set. In this way we cover as much of the input space as possible. We use a test set of $1000$ points, and vary the number of training points to establish the utility of models in both sparse and dense data regimes. We use 1K, 5K and 1M randomly selected training points to evaluate the models in both sparse and dense data regimes.

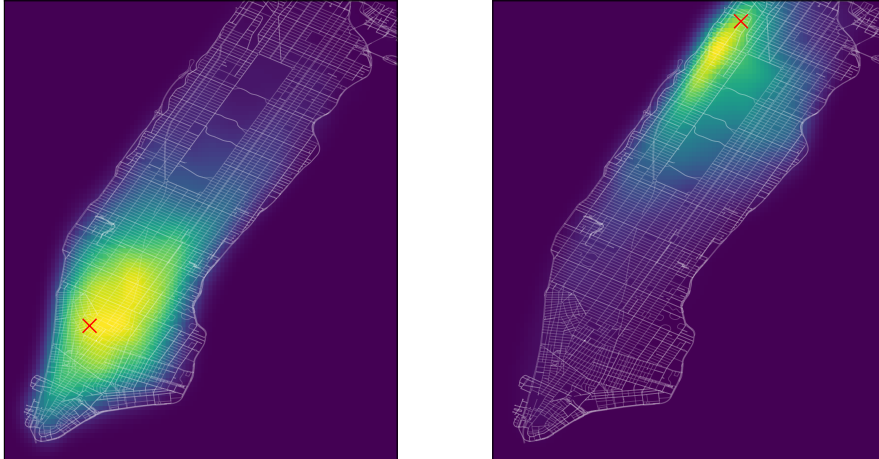

Figure 2: Conditional densities (displayed as heat-maps: yellow means higher probability) of drop-off locations conditioned on the pick-up location (red cross).

Unconditional KDE (U-KDE) ignores the input conditions. It directly models the drop-off locations using Gaussian kernel smoothing. The kernel width is selected with cross-validation. The Conditional KDE (C-KDE) model uses the 50 nearest neighbors in the training data, with kernel width taken from the unconditional model. The table shows that the conditional KDE model performs better than the unconditional KDE model for all conditions. This suggests that the conditioning on the pick-up location and time strongly effects the drop-off location. If the effect of conditioning were slight, the unconditional model should perform better as it has access to all the data.

We also evaluate several MDNs models with differing number of mixing components (MDN-$k$, where $k$ is the number of components), using fully connected neural networks with 3 layers. The MDN model perform poorly except in the large data regime, where the model with the largest number of components is the best performing. The MDN with a large number of components can put mass at localized locations, which for this data is likely to be appropriate as the taxis are confined to streets.

Table 1: NLPP for Manhattan data (lower is better). The models are trained on different dataset sizes.

|        | 1K   | 5K   | 1M    |
|--------|------|------|-------|
| GP-LVM | 2.61 | 2.52 | 2.43  |
| GP     | 2.68 | 2.67 | 2.67  |
| GP-CDE | **2.31** | **2.22** | 2.13  |
| U-KDE  | 2.49 | 2.5  | 2.35  |
| C-KDE  | 2.40 | 2.38 | 2.314 |
| MDN-1  | 2.83 | 2.77 | 2.65  |
| MDN-5  | 2.72 | 2.55 | 2.16  |
| MDN-10 | 3.17 | 2.66 | 2.06  |
| MDN-50 | 5.09 | 3.08 | **1.97** |

We test three GP-based models: our GP-CDE model with 2-dimensional latent variables, and two special cases: one without conditioning (GP-LVM) and one without latent variables (GP). The GP-LVM [26] is our model without the conditioning, and does not perform well on this task as it has not access to the inputs and models all conditions identically. The GP model has no latent variables and independent Gaussian marginals, and so cannot model this data well as the drop-off location is quite strongly non-Gaussian. We added predictive probabilities for all models in Appendix F to illustrate these findings.

The GP-CDE performs best on this dataset for the small data regimes. For the large data case the MDN model is superior. We attribute this to the high density of data when 1 million training points are used. We used a 2D latent space and Gaussian $q(\mathbf{W})$ for the latent variables, with a recognition network amortizing the inference. We use the RBF kernel and use Monte Carlo sampling to evaluate the bound, as described in Section 3.2. For training we use the Adam optimizer with a exponentially decaying learning rate starting at $0.01$ for the hyperparameters, the inducing inputs and on the recognition network parameters. Natural gradient steps of size $0.05$ are used for the GP's variational parameters. Fig. 2 shows the density of our GP-CDE model for two different conditions. Similar figures for the other methods are in Appendix F.

**Few-shot learning** We demonstrate the GP-CDE model for the challenging task of few-shot conditional density estimation on the omniglot dataset. Our task is to obtain a density over the pixels given the class label. We use the training/test split from [19], using all the examples in the training classes and four samples from each of the test classes. The inputs are one-hot encoded (1623 classes) and the outputs are the pixel intensities, which we resize to $28 \times 28$.

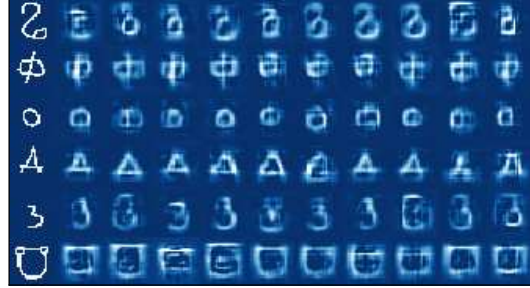

We apply a linear transformation on both the input and output (see Section 3.3). We use a $1623 \times 30$ matrix $\mathbf{A}$ with independent standard-normal priors on each component to project the labels onto a 30-dimensional space. To prevent the model from overfitting, it is important to treat the $\mathbf{A}$ transformation in a Bayesian way and marginalize it.

Figure 3: Sample images for 4-shot learning. Left column is a true (unseen) image, remaining columns are samples from the posterior conditioned on the same label. See supplementary material for further examples.

To correlate the outputs a priori we use a linear transformation $\mathbf{P}$ of the GP outputs, which is equivalent to considering multiple outputs jointly in a single GP with a Kronecker-structured covariance. See Section 3.3. We use 400 GP outputs so $\mathbf{P}$ has shape $400 \times 784$. To initialize $\mathbf{P}$ we use a model of local correlation by using the Matérn-$5/2$ kernel with unit lengthscale on the pixel coordinates and taking the first 400 eigenvectors scaled by the square-root eigenvalues. We then optimize the matrix $\mathbf{P}$ as a hyperparameter of the model. Learning the $\mathbf{P}$ matrix is a form of transfer learning: we update our prior in light of the training classes to make better inference about the few-shot test classes.

We obtain a log-likelihood of $7.2 \times 10^{-2}$ nat/pixel, averaging over the all the test images (659 classes with 16 images per class). We train for 25K iterations with the same training procedure as in the previous experiment. Samples from the posterior on a selection of test classes are shown in Fig. 3. For a larger selection, see Fig. 8 in Appendix F.

**Heteroscedastic noise modeling** We use 10 UCI regression datasets to compare two variants of our CDE model with a standard sparse GP and a CVAE. Since we model a 1D target we consider $\mathbf{w}_n$ to be uni-dimensional, allowing us to use the quadrature method (Section 3.2) to obtain the bound for an analytically optimal $q(\mathbf{w}_n)$. We compare also to an amortized Gaussian approximation to $q(\mathbf{w}_n)$, where we use a three-layer fully connected neural network with $\tanh$ activations for the recognition model. In all three models we use a RBF kernel and 100 inducing points, optimizing for $20K$ iterations using Adam optimizer for the hyperparameters and a natural gradient optimizer with step size 0.1 for the Gaussian variational parameters. The quadrature model use Gauss-Hermite quadrature with 100 points. For the CVAE we use, given the modest size of the UCI datasets, a relatively small encoder and decoder network architecture together with dropout. See Appendix B for details.

Fig. 3 shows the test log-likelihoods using 20-fold cross validation with 10% test splits. We normalize the inputs to have zero mean and unit variance. We see that the quadrature CDE model outperforms the standard GP and CVAE on many of the datasets. The optimal GP-CDE model performs better than the GP-CDE with Gaussian $q(\mathbf{w})$ on all datasets. This can be attributed to three reasons: we impose fewer restrictions on the variational posterior, there is no amortization gap (i.e. the recognition network might not find the optimal parameters [8]), and problems of local optima are likely to be less severe as there are fewer variational parameters to optimize.

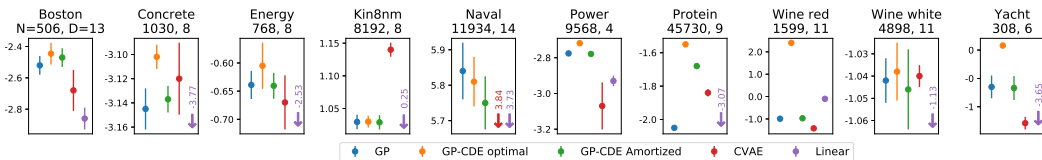

Figure 4: Test log-likelihood of the GP, the optimal GP-CDE, the amortized GP-CDE, the CVAE, and a Linear model on 10 UCI datasets. Higher is better.

**Density estimation of image data** In this experiment we compare the test log-likelihood of the GP-CDE and the CVAE [24] on the MNIST dataset for the image generation task. We train the models with $N = 2, 4, 8, \ldots, 512$ images per class, to test their utility in different data regimes. The model's input is a one-hot encoding of the image label which we concatenate with a 2-dimensional latent variable. We use all $10,000$ test images to calculate the average test log-likelihood, which we estimate using Monte Carlo.

For the CVAE's encoder and decoder network architecture we follow Wu et al. [30] and regularize the network using dropout. Appendix B contains more details on the CVAE's setup. The GP-CDE has the same setup as in the few-shot learning experiment, except that we set the shape of the output mixing matrix $\mathbf{P}$ to $50 \times 784$. We reduce the size of $\mathbf{P}$, compared to the omniglot experiment, as the MNIST digits are relatively easier to model. Since we are considering small datasets in this experiment the role of the mixing matrix becomes more important: it enables the encoding of prior knowledge about the structure in images.

Wu et al. [30] point out that when evaluating test-densities for generative models, the assumed noise variance $\sigma^2$ plays an important role, so for both models we compare two different cases: one with the likelihood variance parameter fixed and one where it is optimized. Table 2 shows that in low-data regimes the highly parametrized CVAE severely overfits to the data and underestimates the variance. The GP-CDE operates much more gracefully in these regimes: it estimates the variance correctly, even for $N = 2$ (where $N$ is the number of *training* points), and the gap between train/test log-likelihood is considerably smaller.

Table 2: Log-likelihoods of the CVAE and GP-CDE models. $N$ is the number of images per class. Higher test log-likelihood is better. See Appendix E for the complete table.

| | CVAE: Fixed $\sigma^2$ | | CVAE: $\sigma^2$ optimized | | | GP-CDE: Fixed $\sigma^2$ | | GP-CDE: $\sigma^2$ optimized | | |
|---|---|---|---|---|---|---|---|---|---|---|
| $N$ | Test | Train | Test | Train | $\sigma^2_{opt}$ | Test | Train | Test | Train | $\sigma^2_{opt}$ |
| 2 | -129.72 | 180.97 | **-1296.63** | 956.39 | **0.01378** | 161.9 | 242.2 | **74.01** | **130.4** | **0.0303** |
| 4 | -60.03 | 178.22 | -759.18 | 956.26 | 0.01364 | 195.2 | 254.2 | 86.59 | 160.3 | 0.0310 |
| 256 | 52.17 | 76.18 | 218.08 | 325.72 | 0.03272 | 606.2 | 545 | 108.1 | 105.4 | 0.0378 |
| 512 | 54.48 | 65.30 | 244.88 | 286.38 | **0.03407** | 606.7 | 512 | 124.2 | 120.7 | 0.0388 |

**Necessity of natural gradients** Natural gradients are a vital component of our approach. We demonstrate this with the simplest possible example modeling a dataset of 100 '1' digits, using an unconditional model with no projection matrices, no mini-batches and no recognition model (i.e. exactly the GP-LVM in Titsias and Lawrence [26]). We compare our natural gradient approach with step size of $0.1$ against using the Adam optimizer (learning rate $0.001$) directly for the variational parameters. We compare also to the analytic solution in Titsias and Lawrence [26], which is possible as we are not using mini-batches. We find that the analytic model and our natural gradient method obtain test log-likelihoods (using all the '1's in the standard testing set) of $1.02$, but the ordinary gradient approach attains a test log-likelihood of only $-0.13$. See Fig. 9 in Appendix F for samples from the latent space, and Fig. 10 for the training curves. We see that the ordinary gradient model cannot find a good solution, even in a large number of iterations, but the natural gradient model performs similarly to the analytic case.

# 6 Conclusion

We presented a model for conditional density estimation with Gaussian processes. Our approach extends prior work in three significant ways. We perform Bayesian linear transformations on both input and output spaces to allow for the modeling of high-dimensional inputs and strongly-coupled outputs. Our model is able to operate in low and high data regimes. Compared with other approaches we have shown that our model does not over-concentrate its density, even with very few data. For inference, we derived an optimal posterior for the latent variable inputs and we demonstrated the usefulness of natural gradients for mini-batched training of GPs with uncertain inputs. These improvements provide us with a more accurate variational approximation, and allow us to scale to larger datasets than were previous possible. We applied the model in different settings across a wide range of dataset sizes and input/output domains, demonstrating its general utility.

## Footnotes

[2]It is common for "$\mathbf{z}$" to denote the latent variables. However, as this letter collides with the notation of inducing inputs in GPs, we will use "$\mathbf{w}$" for the latent variables throughout this paper.

[3]See `https://github.com/hughsalimbeni/bayesian_benchmarks` for the data.

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
