[Supplementary Material]

# A   Notation

Table 3: Nomenclature

| | |
|---|---|
| $n \in \{1 \ldots N\}$ | number of data points $n$ |
| $m \in \{1 \ldots M\}$ | number of inducing variables $m$ |
| $\ell \in \{1 \ldots L\}$ | number of output dimension of the GP $\ell$ |
| $\mathbf{w}_n$ | the latent variable corresponding to data point $n$, $\mathbf{w} \in \mathbb{R}^{D_\mathbf{w}}$ |
| $\mathbf{x}_n$ | $n^{\text{th}}$ input variable, $\mathbf{x} \in \mathbb{R}^{D_\mathbf{x}}$ |
| $\mathbf{y}_n$ | $n^{\text{th}}$ output variable, $\mathbf{y} \in \mathbb{R}^{D_\mathbf{y}}$ |
| $\mathbf{W}$ | matrix collecting all the latent variables $\mathbf{W} = \{\mathbf{w}_n\}_{n=1}^N$, $\mathbf{W} \in \mathbb{R}^{N \times D_\mathbf{w}}$ |
| $\mathbf{X}$ | matrix collecting all the inputs $\mathbf{X} = \{\mathbf{x}_n\}_{n=1}^N$, $\mathbf{X} \in \mathbb{R}^{N \times D_\mathbf{x}}$ |
| $\mathbf{Y}$ | matrix collecting all the observations $\mathbf{Y} = \{\mathbf{y}_n\}_{n=1}^N$, $\mathbf{Y} \in \mathbb{R}^{N \times D_\mathbf{y}}$ |
| $\sigma^2$ | observation noise |
| $\mathbf{f}_\ell$ | function $f_\ell(\cdot)$ evaluated at certain inputs |
| $\boldsymbol{f}(\cdot)$ | collection of GPs, $\{f_\ell(\cdot)\}_{\ell=1}^L$ |
| $k(\cdot, \cdot)$ | prior covariance function of the $d^{\text{th}}$ GP |
| $\mathbf{Z}$ | locations of variational pseudo-inputs |
| $\mathbf{u}_\ell$ | evaluations of the $\ell^{\text{th}}$ GP at the pseudo-inputs: $\mathbf{u}_\ell = \{f_\ell(\mathbf{z}_m)\}_{m=1}^M$. |
| $\mathbf{U}$ | collection: $\mathbf{U} = \{\mathbf{u}_\ell\}_{\ell=1}^L$ |
| $\mathbf{m}_\ell$ | variational posterior mean $\mathbf{u}_\ell$ |
| $\mathbf{S}_\ell$ | variational posterior covariance of $\mathbf{u}_\ell$ |

# B   Network Architectures

In all neural network setups we apply dropout to the output of the hidden layers. The optimal dropout rate is found using grid-search over $\{0.2, 0.1, 0.01, 0.0\}$. We use the Adam optimizer for optimization and perform grid-search over $\{0.01, 0.001, 0.0001\}$ to determine the optimal learning rate. The biases are initialized to zero and the weights using the Xavier distribution [14].

**Heteroscedastic noise modeling on UCI datasets**    Given the modest size of the UCI datasets we choose a relatively small encoder and decoder architecture. The encoder has layers of the following size: $D_\mathbf{x} + D_\mathbf{y}$, 50, 100, 50 and $2 \times D_\mathbf{w}$. The decoder layers have size $D_\mathbf{x} + D_\mathbf{w}$, 10, 50, 50, 10 and $D_\mathbf{y}$. For this experiment we choose a unidimensional latent variable, $D_\mathbf{w} = 1$. The targets of the UCI datasets are also 1D. We use the tanh activation function for all hidden layers and the linear activation function for the final layer.

**Density estimation on MNIST**    In this experiment we follow Wu et al. [29] for the network architecture. The decoder has 5 fully connected layers of size: $D_\mathbf{x} + D_\mathbf{w}$, 64, 256, 256, 256, 1024 and $D_\mathbf{y}$. The tanh activation function is applied to the outputs of the hidden layers and the sigmoid function on the final one. The encoder's fully connected layers have size: $D_\mathbf{x} + D_\mathbf{y}$, 256, 64, $2 \times D_\mathbf{w}$. We use the tanh activation function for the encoder's hidden layers and the linear activation function for the final layer. The inputs are one-hot encoding of the labels, $D_\mathbf{x} = 10$ and $D_\mathbf{y} = 784$.

## C Inference details

In section 3.2 we follow Hensman et al. [15] and use sparse GPs to approximate the full GP. We do this by introducing $M$ inducing or pseudo-inputs $\mathbf{z}_m \in \mathbb{R}^{D_{\mathbf{x}}+D_{\mathbf{w}}}$ and collect them in the matrix $\mathbf{Z} = \{\mathbf{x}_m\}_{m=1}^M$. The inducing outputs $\mathbf{u}_\ell = f_\ell(Z)$ are the function $f_\ell(\cdot)$ evaluated at the inducing inputs. Similarly, we collect the inducing outputs over all dimensions in the matrix $\mathbf{U} = \{\mathbf{u}_\ell\}_{\ell=1}^L$. We assume a Gaussian prior for the inducing outputs $p(\mathbf{u}_\ell) = \mathcal{N}(\mathbf{0}, \mathbf{K})$ and define a posterior distribution of the form $q(\mathbf{u}_\ell) = \mathcal{N}(\mathbf{m}_\ell, \mathbf{S}_\ell)$. The sparse GP framework assumes $\mathbf{u}_\ell$ and $\mathbf{f}_\ell$ to be jointly Gaussian. We can now write $f_\ell(\cdot)$ conditioned on the inducing outputs $\mathbf{u}_\ell$ as

$$f_\ell(\cdot) \,|\, \mathbf{u}_\ell \sim \mathcal{GP}\Big(\mathbf{k}_{\mathrm{Z}}^\top(\cdot)\mathbf{K}_{\mathrm{ZZ}}^{-1}\mathbf{u}_\ell, k(\cdot, \cdot) - \mathbf{k}_{\mathrm{Z}}^\top(\cdot)\mathbf{K}_{\mathrm{ZZ}}^{-1}\mathbf{k}_{\mathrm{Z}}(\cdot)\Big),$$

where $[\mathbf{k}_{\mathrm{Z}}(\cdot)]_m = k(\cdot, \mathbf{z}_m)$ and $[\mathbf{K}_{\mathrm{ZZ}}]_{mm'} = k(\mathbf{z}_m, \mathbf{z}_{m'})$. Marginalizing with respect to $\mathbf{u}_\ell \sim \mathcal{N}(\mathbf{m}_\ell, \mathbf{S}_\ell)$ leads to the variational posterior $q(f_\ell(\cdot))$

$$f_\ell(\cdot) \sim \mathcal{GP}\Big(\mathbf{k}_{\mathrm{Z}}^\top(\cdot)\mathbf{K}_{\mathrm{ZZ}}^{-1}\mathbf{m}_\ell, k(\cdot, \cdot) - \mathbf{k}_{\mathrm{Z}}^\top(\cdot)\mathbf{K}_{\mathrm{ZZ}}^{-1}(\mathbf{K}_{\mathrm{ZZ}} - \mathbf{S}_\ell)\mathbf{K}_{\mathrm{ZZ}}^{-1}\mathbf{k}_{\mathrm{Z}}(\cdot)\Big) =: \mathcal{GP}(\mu_\ell(\cdot), \sigma_\ell^2(\cdot)).$$

As both, the variational posterior over the GP and the likelihood, are Gaussian, $\mathcal{L}_{\mathbf{w}_n}$ (defined in (3)) can be calculated analytically. We start by using the fact that the likelihood factorizes over the output dimension $\mathbb{E}_{q(\boldsymbol{f}(\cdot))}\big[\log p(\mathbf{y}_n \,|\, \boldsymbol{f}(\cdot), \mathbf{x}_n, \mathbf{w}_n)\big] = \sum_\ell \mathbb{E}_{q(f_\ell(\cdot))}\big[\log p(\mathbf{y}_{n,l} \,|\, f_\ell(\cdot), \mathbf{x}_n, \mathbf{w}_n)\big]$. We define a single term of the previous sum as $\mathcal{L}_{\mathbf{w}_n}^\ell$, which can be computed as

$$\mathcal{L}_{\mathbf{w}_n}^\ell = \mathbb{E}_{q(f_\ell(\cdot))}\left[-\tfrac{1}{2}\log(2\pi\sigma^2) - \tfrac{1}{2\sigma^2}\left(y_{n,\ell}^2 + \mathrm{f}_{n,\ell}^2 - 2\,y_{n,\ell}\mathrm{f}_{n,\ell}\right)\right]$$
$$= -\tfrac{1}{2}\log(2\pi\sigma^2) - \tfrac{1}{2\sigma^2}\left(y_{n,\ell}^2 + (\sigma_\ell^2([\mathbf{x}_n, \mathbf{w}_n]) - \mu_\ell^2([\mathbf{x}_n, \mathbf{w}_n])) - 2\,y_{n,\ell}\mu_\ell([\mathbf{x}_n, \mathbf{w}_n])\right).$$

## D Proof of optimality for latent variable posterior

In Section 3.2 we argue that using the optimal free-form distribution $q(\mathbf{w}_n)$ leads to a tighter lower bound. The bound for the optimal and Gaussian form $q(\mathbf{w}_n)$ are, respectively

$$\mathcal{L}_{\mathrm{GAUSS}} = \sum_n \left\{ \mathbb{E}_{q(\mathbf{w}_n)} \mathcal{L}_{\mathbf{w}_n} - \mathrm{KL}\left[p(\mathbf{w}_n)\|p(\mathbf{w}_n)\right] \right\} - \sum_\ell \mathrm{KL}\left[q(\mathbf{u}_\ell)\|p(\mathbf{u}_\ell)\right]$$
$$\mathcal{L}_{\mathrm{FREE}} = \log \mathbb{E}_{p(\mathbf{W})} \left\{ \exp\Big(\sum_n \mathcal{L}_{\mathbf{w}_n} - \sum_\ell \mathrm{KL}\left[q(\mathbf{u}_\ell)\|p(\mathbf{u}_\ell)\right]\Big) \right\}.$$

Starting from $\mathcal{L}_{\mathrm{FREE}}$ and using Jensen's inequality, we get

$$\mathcal{L}_{\mathrm{FREE}} = \log \mathbb{E}_{q(\mathbf{W})} \left\{ \frac{p(\mathbf{W})}{q(\mathbf{W})} \exp\Big(\sum_n \mathcal{L}_{\mathbf{w}_n} - \sum_\ell \mathrm{KL}\left[q(\mathbf{u}_\ell)\|p(\mathbf{u}_\ell)\right]\Big) \right\}$$
$$\geq \sum_n \left\{ \mathbb{E}_{q(\mathbf{w}_n)} \mathcal{L}_{\mathbf{w}_n} - \mathrm{KL}\left[p(\mathbf{w}_n)\|p(\mathbf{w}_n)\right] \right\} - \sum_\ell \mathrm{KL}\left[q(\mathbf{u}_\ell)\|p(\mathbf{u}_\ell)\right].$$

Therefore

$$\log p(\mathbf{Y} \,|\, \mathbf{X}) \geq \mathcal{L}_{\mathrm{FREE}} \geq \mathcal{L}_{\mathrm{GAUSS}}.$$

# E   Density estimation on MNIST: complete table

Table 4: Log-likelihoods of the CVAE and GP-CDE model. $N$ is the number of training images per class. Higher test log-likelihood is better.

| | CVAE | | | | | GP-CDE | | | | |
| | Fixed $\sigma^2$ | | $\sigma^2$ optimized | | | Fixed $\sigma^2$ | | $\sigma^2$ optimized | | |
| $N$ | Test | Train | Test | Train | $\sigma^2_{opt}$ | Test | Train | Test | Train | $\sigma^2_{opt}$ |
|---|---|---|---|---|---|---|---|---|---|---|
| 2 | -129.72 | 180.97 | -1296.63 | 956.39 | 0.01378 | 161.9 | 242.2 | 74.01 | 130.4 | 0.0303 |
| 4 | -60.03 | 178.22 | -759.18 | 956.26 | 0.01364 | 195.2 | 254.2 | 86.59 | 160.3 | 0.0310 |
| 8 | -31.37 | 176.76 | -616.474 | 949.83 | 0.01358 | 234.1 | 269.8 | 124.6 | 168.2 | 0.0312 |
| 16 | -15.50 | 173.34 | -497.15 | 924.11 | 0.01382 | 290.1 | 305.8 | 141.5 | 163.5 | 0.0303 |
| 32 | -1.04 | 161.10 | -382.20 | 800.03 | 0.01577 | 452.7 | 443 | 131.9 | 131.7 | 0.0322 |
| 64 | 18.03 | 130.78 | -227.06 | 530.40 | 0.02348 | 545.2 | 515.7 | 141.2 | 114 | 0.0342 |
| 128 | 41.94 | 97.23 | 76.15 | 399.78 | 0.02959 | 508.3 | 447 | 93.6 | 89.1 | 0.0364 |
| 256 | 52.17 | 76.18 | 218.08 | 325.72 | 0.03272 | 606.2 | 545 | 108.1 | 105.4 | 0.0378 |
| 512 | 54.48 | 65.30 | 244.88 | 286.38 | 0.03407 | 606.7 | 512 | 124.2 | 120.7 | 0.0388 |

# F   Additional Figures

Figure 5: Kernel density estimation models, unconditional (top row), and conditioned on the 50 nearest neighbors.

Figure 6: MDN. Top row: 5 components. Bottom row: 50 components

Figure 7: Five samples from the posterior conditioned on 120 labels randomly chosen from the **training** set. The model saw 20 examples of each image. An example image from the data is shown to the left of each set

Figure 8: Five samples from the posterior conditioned on 120 labels randomly chosen from the **test** set. The model saw 4 examples of each image. An example image from the data is shown to the left of each set

Figure 9: The mean of the GP mapping conditioned on the 1D latent variable, for a dataset of $100$ '1's. Top: the analytic solution of Titsias and Lawrence [26]. Middle: our natural gradient approach, with a step size of $0.1$. Bottom: using the Adam optimizer to optimize the variational parameters $\mathbf{m}$ and $\mathbf{S}$. We see that the ordinary gradient approach is prone to getting 'stuck' in poor local optima, due to the difficulty of optimization.

Figure 10: The training objective for the dataset of $100$ '1's. The three plots show the same three curves with different ranges on the y-axis to highlight the similarities and differences. We can see that natural gradients provide a striking improvement. See Fig. 9 for the samples at the end of training