[Reviews · NeurIPS 2018]

Reviewer 1



This paper designs a model for conditional density estimation. It resembles a VAE architecture, where both x and y are given as inputs to the encoder to produce a latent variable w. W and x are then fed to the decoder to produce p(y|x). However, unlike in VAE, x and y are not single data points, but rather sets and the decoder part uses GPs to output p(y|x). I found the clarity of the paper very low and I wish authors explained the model in Section 3.1. in greater details before jumping to derivations. Figure 1 made me especially confused as I initially thought that the model receives a single datapoint (x,y) just like a VAE. I had to read the GP-LVM paper [20] to realize that x and y are matrices with dimensions (N, Dx) and (N, Dy). Also, I think it’s a too convoluted model for the problem it is trying to solve and I don’t think I would want to use it or implement it myself because of its complexity. I found the experimental results not convincing. It seems that Porto taxi dataset might not be a very good choice considering the fact that a simple unconditional KDE model performs so well. Also, the Omniglot samples in the appendix are a bit disappointing. I would expect train samples to be better compared to the test ones, but I don’t see a lot of a difference. For heteroscedastic noise modeling, the differences between methods are sometimes very small and it’s not clear how significant these results are. With MNIST it was unclear to me if N=2, 4, 256, 512 is the number of training images per class? If so, then it’s no wonder that CVAE overfits with N=2. But what I find very weird is that log-likelihood for GP-CDE is much better on the test set than on the train set in a few cases. Is there an explanation for it? To conclude, I think authors do make an interesting extension of GP-LVMs, however I would not recommend the acceptance of this paper before it is written in a clear way such that there is no need to read the GP-LVM paper to understand what GPs are doing there. Also, the experimental results should be much stronger in my opinion. --------- UPDATE ----------- After reading the rebuttal, I have finally understood how the model works and how the GP is used. I think it's a nice idea, but I'm still convinced that experiments are weak.

Reviewer 2



The paper seems well-written and technically sound. I also believe that the application is very important. Estimating conditional distributions is a very difficult and important task. However, it is not easy to read it and the presentation of used techniques can be improved.

Reviewer 3



Edit after the rebuttal period: I have read the other reviews and the author feedback. My evaluation and rating about this submission will remain unchanged. Summary: The paper considers approximate inference and learning for the Gaussian process latent variables model in which some latent variables are observed. An alternative view of the model considered here is Gaussian process regression in which some dimensions of the inputs are unobserved. Variational inference based on pseudo-points is used. The posterior over the pseudo-points are approximately integrated out by quadrature or approximated by a Gaussian parameterised by a recognition model. Natural gradients and linear input and output warpings are also considered. Extensive experiments on density estimation and regression were provided to support the proposed inference scheme. Comments: The contribution of this work is, in my opinion, a well-executed framework for inference and learning in Gaussian process latent variable models with some latent variables being known beforehand. I think the contributions listed at the end of section 1 are incremental, as i. linear input and output warpings have been considered before in multiple contexts, for example: multi-task learning, ii. natural gradients for the variational approximation over the non-linear function have been considered before, e.g. Hensman et al's GP for big data paper. In addition, recognition models have been used before to parameterise the variational distribution in the GP latent variable model, and similar models have been considered before -- see the list at the end of the review. Having said this, this submission is well-structured and is an interesting and, perhaps, non-trivial combination of all these things that seems to work well. re analytically optimal q(w): I agree this approach could be useful for low-dimensional w and there will be less variational parameters to deal with [and this could perform well as in the regression experiment considered], but I'm pedantically unclear about the bound being tighter here. The gaps between the bound and the true marginal likelihood are: in the free-form q(w) case: log \int dw p(w) \exp [ KL(q(f) || p(f|w, x, y)) ] in the fixed-form case: KL( q(f)q(w) || p(f, w|x, y) ) Is this true that the gap in the first case above is always smaller than that in the second case? Some questions about the experiments: + linear model baselines should be included in the density modelling tasks + taxi trip density estimation: the test set has 4 folds with 50 taxi trips each -- this seems rather small compared to the whole dataset of 1.5 mil trips. + few shot learning on omniglot: the generated images are not very crisp. Is there some quantitative or qualitative comparison to alternative models here, e.g. VAE? + regression task: this seems like the only task the proposed model + inference technique have the competitive edge on average. Is this because the number of latent/input dimensions is small and no input + output warpings are used? The additional variational approximation for the input warping could make things worse in other tasks. + MNIST density estimation: why not use the full binarised MNIST as often considered in generative modelling literature? It is hard to judge the number provided table 2 as they are very different to what often reported. The Gaussian likelihood could be straightforwardly replaced by another likelihood for binary outputs, right? I'm also slightly puzzled by table 2 and the full table in the appendix: why is the first row for CVAE: sigma^2 optimised made bold? For fixed variances, GP-CDE is like 100-500 nats better than CVAE but for optimised variance, GP-CDE is, disappointedly, way worse compared to CVAE? Could you please enlighten me on this: perhaps the capacity of the GP-CDE model is not great, or is that limited by the inference scheme? Would CVAE + early stopping work much better than GP-CDE overall then? Clarity: The paper is very well-written and the experiments seem to be well thought out. Perhaps the following works should be mentioned and discussed in details, as they are closely related to this submission: Lawrence, Neil D., and Joaquin Quiñonero-Candela. "Local distance preservation in the GP-LVM through back constraints", ICML 2006 -- this was briefly mentioned, but the idea of using recognition models for GP latent variables was first introduced in this paper, albeit only for MAP. Thang Bui and Richard Turner, "Stochastic variational inference for Gaussian process latent variable models using back constraints", NIPS workshop on Black Box Learning and Inference, 2015 --- this paper uses recognition models for GP latent variable models Zhenwen Dai, Andreas Damianou, Javier González, Neil Lawrence, "Variational Auto-encoded Deep Gaussian Processes", ICLR 2016 -- using structured recognition models for deep GPs. A. Damianou and N. Lawrence, "Semi-described and semi-supervised learning with Gaussian processes", UAI 2015 --- This paper is very related to the submission here, though the set up is slightly different: some inputs are observed and some inputs are unobserved/missing and treated as latent variables. Depeweg S., Hernández-Lobato J. M., Doshi-Velez F. and Udluft S., "Learning and Policy Search in Stochastic Dynamical Systems with Bayesian Neural Networks", ICLR 2017 --- the set up in this paper is very similar to what proposed here, except that Bayesian neural networks were considered instead of GPs